# Timing-Optimised 3D Silicon Sensor with Columnar Electrode Geometry

**DOI:** 10.3390/s25030926

**Published:** 2025-02-04

**Authors:** Angelo Loi, Adriano Lai, Jixing Ye, Gian-Franco Dalla Betta

**Affiliations:** 1Istituto Nazionale di Fisica Nucleare (INFN), Sezione Cagliari, Dipartimento di Fisica, Strada Provinciale per Sestu, km 0,700, 09042 Monserrato, Italy; 2Dipartimento di Ingegneria Insdustriale, Università degli Studi di Trento e TIFPA, Via Sommarive 9, 38123 Povo, Italy

**Keywords:** 3D silicon detectors, parallel computing, 4D tracking, fast timing, LHCb, VELO, HL-LHC

## Abstract

Among various silicon sensor technologies, 3D silicon sensors demonstrate significant potential for applications requiring exceptional radiation hardness and intrinsic high time resolutions. Silicon pixel sensors with columnar-type electrodes are already operational within the ATLAS experiment, serving in the previous Inner B-Layer (IBL) and the upcoming Inner Tracking (ITk) detectors. Concurrently, advancements driven by the next-generation LHCb VELO detector have led to the development of fast-timing 3D trench sensors within the INFN TimeSPOT project, achieving intrinsic time resolutions close to 10 ps. Remarkably, this performance is sustained even under irradiation levels far exceeding the expected limits for High Luminosity LHC operations. Despite these advantages, 3D trench sensors face challenges related to fabrication, as their production yields remain lower than those of the well-established columnar-type sensors. This highlights the necessity of designing a timing-optimized 3D sensor that leverages the robustness of a columnar electrode fabrication while achieving an intrinsic time resolution as close as possible to the trench-based designs. The design study addressed in this paper aimed to computationally compare the already designed and characterised TimeSPOT 3D trench sensor with alternative columnar electrode-based geometries, focusing particularly on configurations that approximate trench electrodes using parallel-oriented columnar designs. Different geometries and pixel sizes were designed, simulated, and compared. This work presents the entire design and selection effort as well as the preliminary layout of the selected pixel geometries, which are set to feature in FBK’s upcoming production run in 2025.

## 1. Introduction

Physics experiments at colliders are gearing up to tackle the challenge of high intensity. This begins with the high-luminosity upgrade to the LHC but will continue and increase with subsequent developments at the FCC. Furthermore, the ability to sustain increasing intensities could enable access to innovative experimental techniques, such as particle tagging and even particle identification at the vertex detector level.

To achieve next-generation tracking techniques, it is essential to develop sensors and electronics with specific features: high radiation resistance and high spatial and temporal resolution. The need for simultaneous position and time measurements at the single-pixel level has given this experimental technique the name of *4D tracking*.

In its developments dedicated to the high luminosity upgrade, known as Upgrade 2 (or U2), the LHCb experiment is the first to define technical requirements for 4D tracking that could allow the vertex detector to maintain its detection and reconstruction efficiencies, despite the increased luminosity and the resulting event pile-up in the detector [1].

These requirements correspond to a due resistance to particle fluences on the sensors of 5 × 10^16^ 1-MeV n_*eq*_/cm^2^, a spatial resolution of 10 µm (achievable with pixel dimensions of approximately 50 µm), and a temporal resolution no worse than 50 ps at system level, which translates to 30 ps or better of a sensor time resolution throughout its operational lifetime, i.e., also after the expected high levels of radiation exposure [2].

For the expected increment in radiation damage, among possible sensor technologies able to cope with these requirements, 3D silicon pixels have attracted great interest in recent years. First introduced by S. Parker in 1997 [3], 3D sensors feature vertical electrodes penetrating all through the substrate. In this way, the inter-electrode distance is decoupled from the substrate thickness and can be made small enough (tens of micrometers) by a layout, so that a relatively small bias voltage can be adopted and charge trapping effects can be mitigated [4]. Extreme radiation hardness can, thus, be achieved with relatively low power dissipation. Owing to these advantages, 3D sensors were first used in the ATLAS Insertable B-Layer [5] and were later chosen to equip the innermost layer of the ATLAS Inner Tracker at the high-luminosity LHC [6].

Three-dimensional pixel sensors are also inherently fast devices, but this property has not been fully exploited yet. Apart from [7], where results from early test structures and hints for sensor optimization were reported, until recently, no other study has been devoted to exploring the potential of 3D sensors for timing. In [8], the timing properties of a 50 × 50 μm^2^ 3D single-pixel test structure made at IMB-CNM (Barcelona, Spain) with a double-sided process on a 285 μm thick substrate was tested with a fast discrete readout channel, showing a timing resolution of 30 ps at 150 V bias and −20 °C. The same type of test structure was also later tested after irradiation up to 1.0 × 10^16^ 1-MeV n_*eq*_/cm^2^, showing a timing resolution lower than 50 ps at 150 V bias and −20 °C [9]. More recently, another study involving the same type of test structure made at IMB-CNM in two versions of 230 μm (double-sided) and 150 μm (single sided) active thicknesses was reported, showing a timing resolution of 25 ps at −20 °C after irradiation to a fluence of 5.0 × 10^16^ 1-MeV n_*eq*_/cm^2^ [10].

Although these results are very good, they are ultimately limited by the electric field and weighting field spatial non uniformities, which are typical of 3D sensors with columnar electrodes [3]. In this respect, as proposed in [7], better results can be obtained by replacing columnar electrodes with trenched electrodes at the expense of a larger capacitance and technology complication. This was indeed demonstrated in the INFN TimeSPOT Project [11], which successfully developed 3D-trenched sensors in collaboration with FBK (Trento, Italy). Devices were fabricated on Silicon-Silicon Direct Wafer Bonded substrates of a 150 μm active thickness using a single-sided technology [12] that was adapted from the process developed for the 3D column pixels of ATLAS ITk [13]. Beam test results from 55 × 55 μm^2^ 3D single-pixel test structures with fast discrete readout channels demonstrated a time resolution well below 20 ps at room temperature, in good agreement with simulations [14]. Successive test beam campaigns with improved electronics based on a transimpedance amplifier [15] achieved even better results, of the order of 10 ps, maintaining this performance even after irradiation up to 1.0 × 10^17^ 1-MeV n_*eq*_/cm^2^ [16], only requiring a slight increase in the bias voltage with respect to the pre-irradiation case and operation in the temperature range from −40 °C to −20 °C using dry ice.

The intrinsic timing resolution of 3D-trenched sensors is really outstanding, but it might not be maintained in pixel implementations, due to the power constraints in the readout chip. The pixel capacitance is a possible concern, due to its impact on the noise. Moreover, the fabrication technology for 3D trench pixels is very complex. Also, 3D column pixel technology is itself quite difficult but has now reached its maturity and is compatible with medium-volume productions, as confirmed by the successful fabrication at FBK of several hundreds of large-size sensors (4 cm^2^) for the ATLAS ITk. On the contrary, 3D trench pixel technology is still in an early development stage and the fabrication yield is not yet adequate for the production of large-size pixel sensors. The question then arises as to whether 3D column pixels could be optimized to provide comparable performance to 3D trench pixels while easing the manufacturability. The goal of this paper is to investigate and optimize 3D column designs with different cell sizes and column arrangements, aimed at the implementation of a dedicated fabrication batch at FBK. Section 2 introduces the proposed sensor geometries and dimensions, along with the initial layout designs. Section 3 details the simulation workflow used for the design and time-domain simulations aimed at predicting the performance of the proposed geometries. Section 4 presents the results of static simulations and the initial selection process based on capacitance analysis, while Section 5 focuses on the outcomes of time-domain simulations performed using TCAD-TCoDe with advanced parallel computing.

## 2. Device Description

This study aimed to evaluate, through simulation, alternative 3D electrode geometries based on columns instead of trench-electrodes as a substitute for the 3D trench design. The proposed geometries, summarized in Table 1 and sketched in Figure 1, include configurations with one electrode (1E), two electrodes (2E), and three electrodes (3E). Each geometry is examined with pixel pitches of 45 µm, 50 µm, and 55 µm. The performance of these designs is compared with the TimeSPOT 3D trench pixel, which is already characterised over all possible aspects, ranging from simulation to beam test data.

Each device is intended to be fabricated using the same technology developed at FBK for the ATLAS ITk production [13]. The sensor wafer has a total thickness of 650 µm, with 150 µm constituting the active bulk, while the remaining thickness serves as a mechanical support that can be thinned away after fabrication if necessary. Due to the aspect ratio of the fabrication process of the columns, the bias column has a diameter of 6 µm and extends through the entire active thickness of the wafer. Meanwhile, the readout electrode is 5 µm in diameter and penetrates to a depth of 120 µm within the wafer, ensuring a sufficiently high breakdown voltage to operate all devices well above 100 V.

All the 1E and 2E geometries considered in this work will be implemented in a fabrication batch to be launched at FBK at the beginning of 2025. The batch will be processed using stepper lithography in order to profit from the very good definition of critical layout details. The reticle will include pixel arrays of different sizes and pitches compatible with existing and future readout chips [17], as well as test structures (e.g., single pixels, strips, diodes, etc.) As an example, Figure 2 shows the layout of pixels with 45 µm pitch, along with a schematic cross-section of the devices.

## 3. Simulation Flow

The primary simulation tool employed in this study was Synopsys Sentaurus TCAD [18], specifically its modules Sentaurus Device (SDevice) and Sentaurus Structure Editor (SDE). SDE was utilised to design the pixel geometry and its structural properties, while SDevice was used to conduct a detailed simulation of the static characteristics of the geometries. This includes the calculation of their weighting field, the generation of the Ramo Map, and the simulation of the capacitance of the pixel. The capacitance analysis was performed through mixed-signal simulations within the SDevice module, with an accurate modelling of the electrostatic behaviour of the pixel sensor.

For the study of the transient properties, a more sophisticated simulation workflow was established, integrating three distinct simulation tools. Synopsys TCAD provides critical physical parameters of the sensor, including its electric field, carrier mobility, and weighting field. A dedicated GEANT4 [19] simulation simulates the energy release of high-energy particles passing through the silicon sensor. Sensor and interaction data are then included within the TCoDe [20,21], which performs charge drift, diffusion, and current signal generation.

## 4. Static Simulation

### 4.1. Weighting Field

The study of the weighting field was conducted using Synopsys Sentaurus TCAD, applying weighting potentials to the readout electrode according to the Ramo Theorem [22]. When comparing the 1E, 2E, 3E, and 3D trench geometries, a distinct trend emerges: the 1E design generates a weighting field, which is weaker and extends farther outside its own perimeter compared to all other geometries (Figure 3, relevant to the 55 µm pitch). This finding suggests that the 1E design is more prone to charge-sharing effects in all directions than the 2E, 3E, and 3D trench designs due to their more closed electrode geometry. The 2E design already demonstrates a far better confinement of the weighting field within its active volume, while the TimeSPOT 3D trench offers the best overall performance. Its continuous trenches effectively restrict the weighting field to a single direction associated with the bias line. Comparing weighting fields based on pixel size, the behaviour does not change except for the amplitude, which increases with a smaller pixel pitch.

### 4.2. Capacitance

The total capacitance of the pixel consists of two main contributions: one from the bulk, depending on the pixel geometry and size, and another one from the surface, depending on the details of the different layers involved (e.g., p-spray, oxide, poly-silicon, metallization, etc.). While the bulk contribution to the capacitance can vary significantly between devices, the surface contribution is nearly identical across them and typically adds an additional 10–20 fF to the total capacitance. The bulk capacitance was simulated using the Sentaurus SDevice in a small-signal simulation mode. In Figure 4, it is evident that the smallest capacitance corresponds to the 1E design with a capacitance close to 35 fF. The 2E design exhibits a capacitance approximately twice as large, close to 75 fF, while the 3E geometry slightly surpasses that of the 3D trench. Another observation is that the capacitance of the same geometries shows minimal variation with different pixel sizes, making the number of electrodes the primary factor to consider, excluding sensor thickness. At this phase of the study, the 3E design was excluded from further computational studies, as its increased capacitance, which basically is the same as the 3D trench, does not justify the additional technological and fabrication effort required to produce a similar device with timing performances similar of those of the 3D trench but with a slightly higher capacitance.

### 4.3. Ramo Maps

The Ramo Map approach has been developed specifically to study the current that a single charge carrier can induce on a readout electrode once the bias voltage is set [20]. This allows us to estimate the uniformity of the response of the sensor. The Ramo Map is computed using the Ramo Theorem by combining the charge drift velocity maps and the weighting field maps generated by TCAD. This is performed on a point-by-point basis, with the specific charge of the carrier of interest selected to calculate the induced current. In this study, the 3E geometry was excluded due to its intrinsic large capacitance, reducing the study to the 3D trench, 1E, and 2E designs, all of them utilising a 150 µm thick active silicon layer. Figure 5 presents the electron Ramo Maps for all the geometries under investigation, computed at a bias voltage of 100 V. From the figure, it is evident that the 3D trench geometry offers a significant advantage over all columnar-based electrode designs. This is attributed to its parallel-wall electrode configuration, which ensures high and uniform coverage of both the electric field and the weighting field.

In contrast, the columnar electrode-based geometries exhibit a specific trend related to pixel size: as the pixel size decreases, the induced current increases due to the larger electric field and, therefore, faster drift velocity. Notably, the 2E geometry with a pitch of 45 µm demonstrates the highest current induction among all the designs studied, positioning it as a strong candidate for the timing-optimized 3D column sensor.

## 5. Transient Simulation

The time-domain simulation is based on a custom workflow developed within the TimeSPOT project combining commercial and in-house-developed tools. TCAD is a key component, providing the 3D structures and maps with the physical properties of the sensor required to compute charge drift, diffusion, and current induction via the Ramo theorem. To optimize computational efficiency, only one-quarter of the pixel is modelled and electrically simulated. A tensorial mesh grid is implemented instead of a traditional Delaunay-type mesh [23]. TCAD operates by default using the Delaunay mesh grid, which is a type of triangular or tetrahedral mesh optimized for computational geometry, ensuring that no vertex lies inside the circumsphere of any tetrahedron, which improves numerical stability and element quality. In contrast, a tensorial mesh (also known as structured mesh) is based on a regular grid of elements, often aligned with coordinate axes, which simplifies computation but may be less flexible for complex geometries. Delaunay meshes adapt well to irregular domains, while tensorial meshes excel in regular, rectangular regions. For this application the tensorial mesh is chosen over the Delaunay because during computation it allows the TCoDe algorithm to efficiently retrieve the physical parameters needed for charge transport calculations at each time step. The generated models are simulated using quasi-stationary simulation on SDevice with a reference bias voltage of 100 V used for the comparison. After the simulation, the electric field, carrier mobility, and weighting field are extracted as datafiles and post-processed to reconstruct the full pixel matrix using the symmetry of the pixel geometry. This approach is used to generate the 3 × 3 pixel matrix, in which the test pixel is located at the centre (Figure 6). A separate model for the weighting field incorporates the boundary condition where the adjacent pixels are set to a zero weighting potential relative to the test pixel.

Particle–matter interactions were simulated using GEANT4. The setup models a silicon block with dimensions 165 × 165 × 150 µm^3^, which replicates the entire volume of a 3 × 3 pixel matrix of a 55 µm pitch sensor. The target block is positioned in front of the primary particle source, a 180 GeV positive pion beam, to replicate test beam conditions at the CERN SPS. Only the central pixel is targeted. Three tilt angles were considered for the study: 0°, 10°, and 20°. For each angle, 15,000 events were simulated, only varying randomly the initial particle coordinates along the x- and y-axes with a dispersion corresponding to the largest pixel pitch (55 µm), which are uniformly distributed over a generation surface parallel to the silicon block. The size of the generation surface takes into account the potential track of the particle within the entire pixel under study (Figure 7). This means that, if tilted, the signals generated by the pixel under simulation might have passed through the surface of the neighbouring pixel, which is presented in the next section. The dataset generated for each angulation is the same used for all geometries and sizes. This allows us to compare one-by-one the effects of the same family of energy deposits on different geometries and pixel sizes by accepting that smaller pitches will have a slightly reduced statistic due to their smaller active cross section.

The data for each hit are subsequently extracted and input into TCoDe, which processes the particle track and deposited energy to generate the corresponding spatial distribution of electron–hole pairs. Such distribution is the initial condition of the simulated event in the transient simulation. TCoDe transient simulations were executed on an NVIDIA A100 GPU, achieving an average runtime of 730 ms per transient simulation. Processing a complete dataset of 15,000 events required approximately 3 h. The transient duration was set to 2000 ps with a time step of 1 ps. TCoDe outputs the results of each event, providing the impact point on the sensor surface, the generated current signal, the contributions from primary and secondary ionizing particles, and the total collected charge.

### 5.1. Analysis and Results

Each dataset was analysed based on key quantities, particularly the collected charge and the charge collection time (CCT). CCT is a key quantity that determines the intrinsic time resolution of a silicon detector. As explained in [15], by using a dedicated fast analogue readout based on a trans-impedance amplifier (TIA), the intrinsic time resolution is strongly dependent on the charge collection time distribution, improving performance compared to a more classical charge sensitive amplifier. This allows us, in the first phase, to estimate the potential time resolution of a complete detection chain (sensor + analogue front end) by only analysing the CCT distribution of the sensor.

#### 5.1.1. Charge Collection Time

For this study, the CCT was evaluated at various thresholds, which are 25%, 50%, 75%, and 100%, to display how efficiently the sensor collects the majority of the charge. A threshold in charge at 1000 electron–hole pairs was applied as the minimum amount of charge detectable from a potential readout ASIC. Figure 8 shows the average trend observed among all geometries with respect to the TimeSPOT 3D trench sensor.

Several pieces of information can be derived from the presented graph. First, it confirms that a smaller pixel pitch enables faster average charge collection due to a shorter drift path. Secondly, the 2E design demonstrates approximately two times faster charge collection than an equivalent-pitch 1E geometry, which is to be ascribed to the larger number of columns and their more efficient distribution within the active volume. The best results are observed for the 2E geometry with a 45 µm pitch, closely matching the performance of the TimeSPOT 3D trench sensor. Additionally, all geometries shown improve the charge collection up to 75% of the total charge. Beyond this point, the collection time performance decreases, which can be attributed to two specific geometrical aspects shared among all geometries. First of all, all geometries present low electric field volumes sited between electrodes at the same potential. The electric field in those area drops close to zero and the charge drift becomes slow, causing tails on the CCT distribution. The second factor is related to the length of the readout electrodes, which are 120 µm deep, covering 80% of the total thickness of the sensor. Within this active volume, charge collection is more efficient in the region where readout electrodes and bias electrodes are facing each other, minimising drift distance. In contrast, the remaining 20% is lacking this geometrical configuration, increasing the minimum drift distance to 30 µm instead of 23 µm. At the same time, the larger inter-electrode distance in that region reduces the electric field amplitude, which worsens drift velocity and, therefore, charge collection efficiency, slowing it down compared to the upper regions. The TimeSPOT 3D trench sensor also exhibits a similar behaviour, especially the second one, but the parallel wall configuration reduces the number of low electric field areas within the active bulk of the sensor.

The comparison of the dispersion of the charge collection time distributions reveals a trend similar to that observed with the average charge collection time. As expected, the 1E geometries exhibit broader CCT distributions, indicative of a wider spread of the CCT. In contrast, the 2E geometry demonstrates narrower distributions, reflecting more uniform charge collection, with the best performance at a 45 µm pitch (Figure 9). This behaviour can be attributed to the same reasons that influence the average CCT. A possible way to reduce this “tail effect” can be achieved by increasing the length of the readout electrode to the extent that the distance to the bottom of the sensor corresponds to the inter-electrode distance. This could reduce the low electric field volumes to those present between electrodes at the same potential and the corner regions of the bottom volume of the sensor. This approach will be considered for the incoming production batch.

#### 5.1.2. Collected Charge

Regarding the collected charge, tilting significantly influences the quantity within the sensor’s active volume. Figure 10 presents the comparison between the collected charge distribution and its spatial distribution within the surface of the sensor matrix crossed by the particles during the GEANT4 simulation. At 0° (Figure 10A1,B1), the sensor’s geometry is visible, with columns and trenches forming dead volumes with small charge deposits that reduce detection efficiency. The charge distribution aligns with the typical behaviour of a Landau distribution. Tilting the sensor to 10° (Figure 10A2,B2) and then to 20° (frames Figure 10A3,B3) increases detection efficiency. This is due to the fact that an inclined sensor with respect to the direction of propagation of the incoming MIP prevents the particle from passing entirely through the electrodes, which act as dead volumes. However, this also reduces the charge released per pixel per event, redistributing it among two or more pixels at minimum for the same event, potentially reducing the frontend performance and final time resolution of the detection chain.

From the tilted maps, it is possible to see the effects of the GEANT4 simulation performed in Figure 7. The maps and corresponding charge distribution display all events generated within the GEANT4 setup that triggered a signal response above 1000 electron–hole pairs on the simulated pixel (delimited by the yellow perimeter), including not only the events that crossed the main pixel but also all the events that crossed the pixel positioned below.

In fact, given a certain sensor thickness and pitch, a maximum tilting angle can be determined by applying the arctangent of the ratio of pixel pitch to thickness. At this angle, the total energy released by the particle is distributed among two pixels with ratios of at worse 50% per pixel. Above that angle, the charge starts to be distributed over three or more pixels, further reducing the overall time resolution of the entire detection chain. Below that angle, most of the charge released per event stays within a single pixel, without considerable loss in performance.

For example, in the case of a 150 µm thick silicon sensor, such as the scheduled batch presented within this work, the most probable value of energy release is approximately 2 fC. Analogue frontends in integrated readout chips currently under development are designed to provide detection of 1 fC of charge with a time resolution of 40 ps or less [24]. This means that for sensors with a 55 µm pitch, the maximum angle is 20°; for 50 µm, it is 18.43°, and it is 16.7° for a 45 µm pitch. In Figure 10, this angle is displayed showing the difference between the 2E geometry and the 3D trench (Figure 10A3,B3). For the 20° tilting, the map displayed for the 55 µm pitch 3D trench has a symmetric distribution of charge around the border between two pixels, while for the 45 µm pitch, this is not the case, detecting most events with the largest charge release that are crossing the lower pixel.

## 6. Conclusions

An intense design and simulation campaign was carried out to develop a timing-optimized 3D sensor geometry based on columnar electrodes instead of trenched ones. A relative comparison between the 1E and 2E geometries to the reference TimeSPOT 3D trench sensor was conducted and evaluated. The 3D trench sensor served as a reliable benchmark, as its simulation results were validated with experimental results collected on different test beam campaigns during the last five years. This established validation framework ensures the robustness of the comparative analysis, providing a solid basis for evaluating the performance of potential candidate geometries. Based on this comparison, the 2E geometry at 45 µm emerges as the promising candidate for production in the upcoming batch, showing a lower capacitance by maintaining a relatively fast average charge collection time with a very narrow distribution, comparable to the reference TimeSPOT 3D trench. Considering beam test results achieved with the TimeSPOT 3D trench pixel [16] and 1E geometries with a 50 µm pixel size [8] and the correlation between the time resolution and charge collection time distribution [15], the intrinsic time resolution of the 2E geometry with a 45 µm pixel size should be potentially located within the lower part of the interval between 10 ps and 45 ps. 

## Figures and Tables

**Figure 1 sensors-25-00926-f001:**
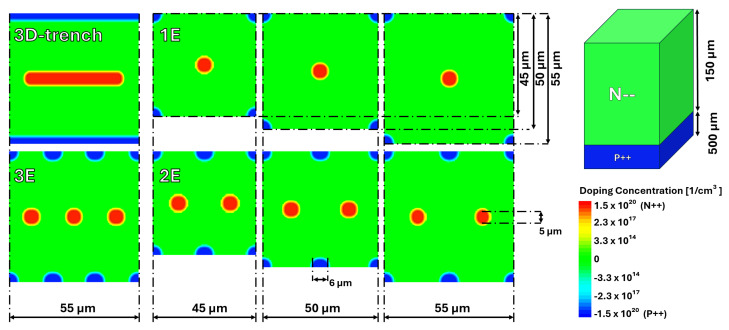
Sensor geometries involved in the study.

**Figure 2 sensors-25-00926-f002:**
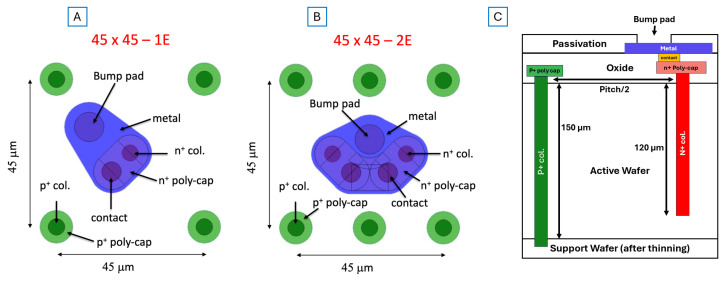
Layout of 3D column pixels with 45 µm pitch: (**A**) 1E, (**B**) 2E, and (**C**) cross section showing column structure (not to scale).

**Figure 3 sensors-25-00926-f003:**
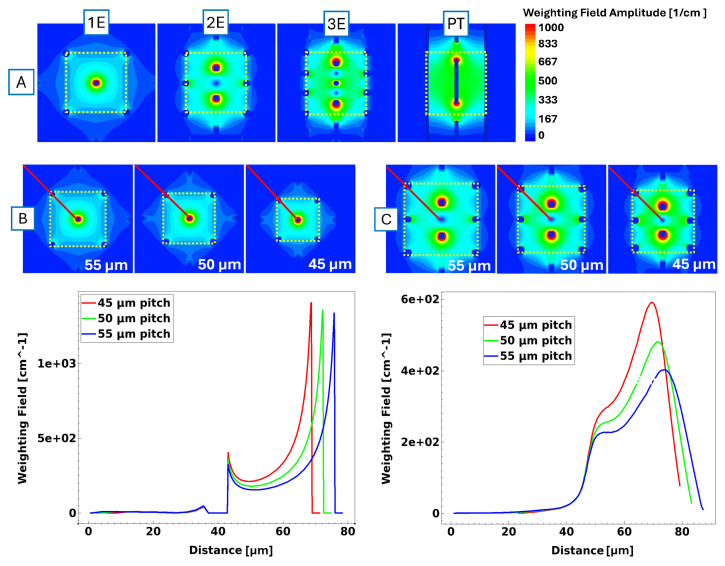
(**A**) Weighting field for all designed geometries at a 55 µm pitch. Pictures to scale. The yellow line shows the perimeter of the pixel under simulation. (**B**,**C**) show the weighting field behaviour for the 1E and 2E designs, respectively, at different pixel sizes. The red line indicates the probeline used to analyse the weighting field amplitude from the centre towards the outside, compared at the respective plot on the right.

**Figure 4 sensors-25-00926-f004:**
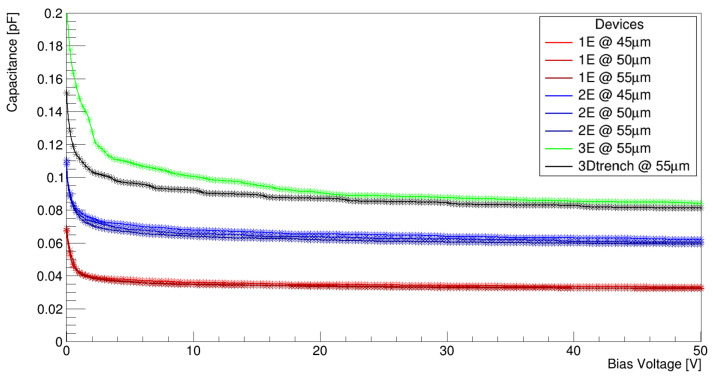
Bulk capacitance for all designed geometries.

**Figure 5 sensors-25-00926-f005:**
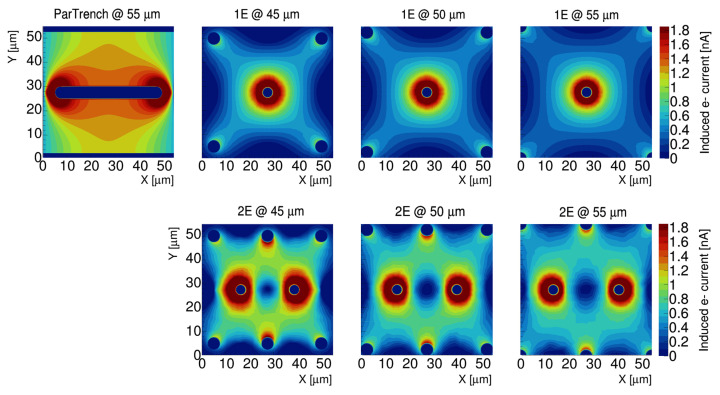
Overview of electron Ramo Maps for different sensor geometries and sizes. From the different maps, it is evident how the 2E geometry at a 45 µm pitch is second only to the TimeSPOT 3D trench in terms of current induction.

**Figure 6 sensors-25-00926-f006:**
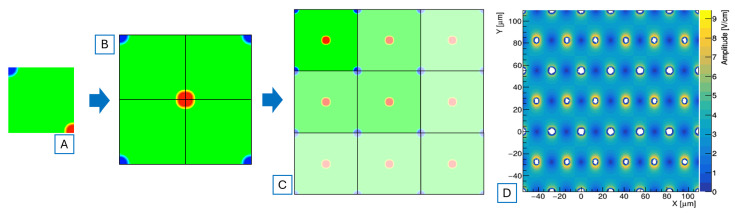
Generation of large pixel structures starting from a single TCAD model. (**A**) Section of the starting model. (**B**) Single pixel rebuilt using its symmetry. (**C**) Replication of a single pixel structure as a 3 × 3 matrix. (**D**) Electric field map (amplitude) of a 2E, 3 × 3 pixel matrix.

**Figure 7 sensors-25-00926-f007:**
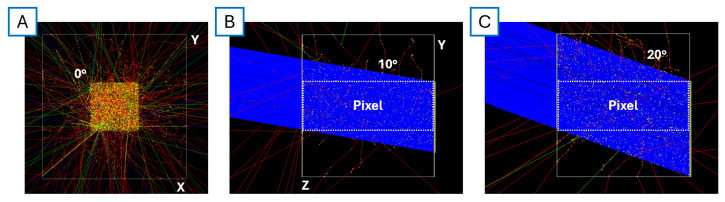
GEANT4 setup developed for the study. (**A**) Front view of the target when shot with particles at 0°. (**B**,**C**) show lateral views of the target with 10° and 20° particles, respectively.

**Figure 8 sensors-25-00926-f008:**
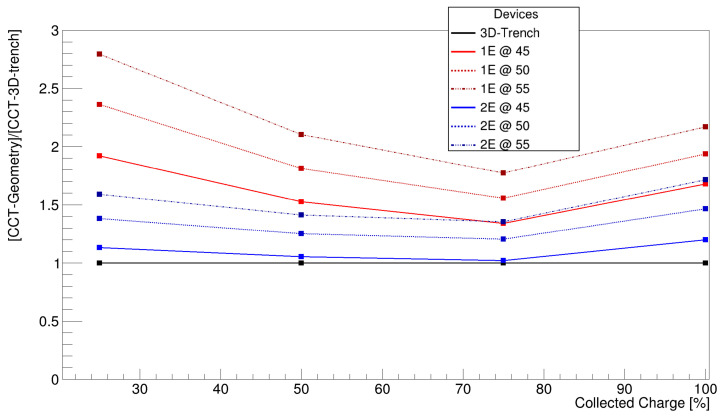
Relative average charge collection time compared to TimeSPOT 3D trench at different fractions of the collected charge.

**Figure 9 sensors-25-00926-f009:**
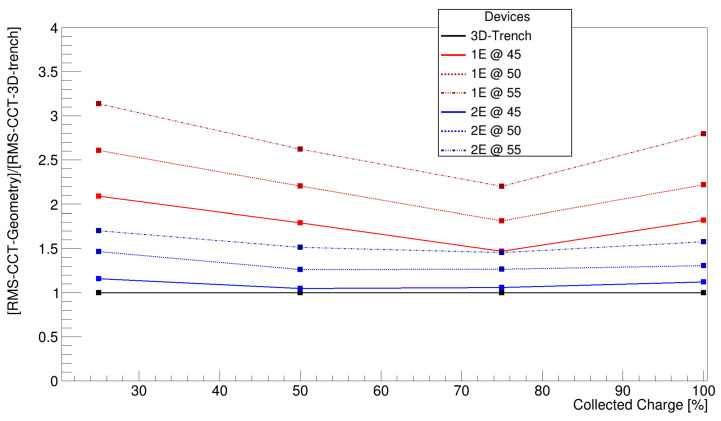
Relative root mean square of the charge collection time distributions of 1E and 2E geometries compared to 3D trench.

**Figure 10 sensors-25-00926-f010:**
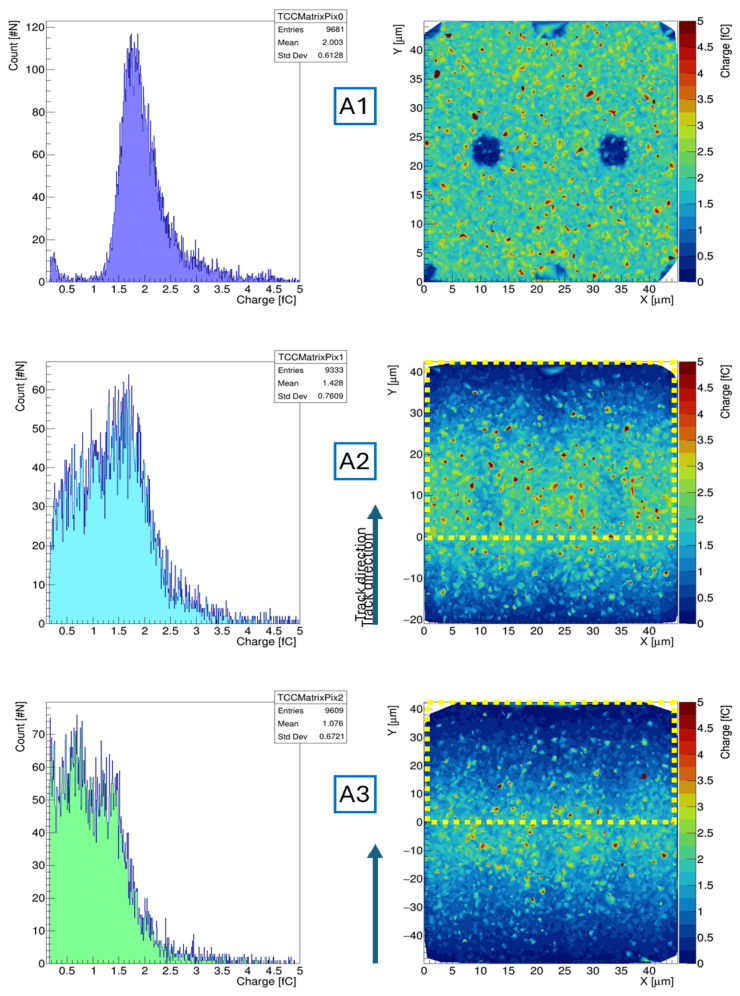
Comparison between the 2E geometry (**A**) at a 45 µm size and the TimeSPOT 3D trench at 55 µm (**B**). Angulation at 0° is **A1** and **B2**, at 10° is **A2** and **B2**, and at 20° is **A3** and **B2**. We pay attention to the different scales on the y-axis. An important aspect can be seen by comparing the charge distributions at the same tilting but different pixel sizes. In the case of the 10° tilting, the 45 µm loses most of the core Landau distribution; meanwhile, the 55 µm pitch still presents an important charge peak. While on pictures **A1** and **B1** the entire pixel is already displayed, a yellow perimeter in used in sequence (**A2**,**B2**) and (**A3**,**B3**) to highlight the effective position of the pixel under simulation.

**Table 1 sensors-25-00926-t001:** Designed sensor geometries and pitch.

45 µm	50 µm	55 µm
1E	1E	1E
2E	2E	2E
not included ^2^	not included ^2^	3E
not included ^2^	not included ^2^	ParTrench ^1^

^1^ As reference device. ^2^ Due to technological limitations of the pad size.

## Data Availability

Data is available by contacting corresponding author who will provide within maximum 10 working days access to shared data folder.

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
