# Peer review of "Timing-Optimised 3D Silicon Sensor with Columnar Electrode Geometry"

_sensors, 2025, doi:10.3390/s25030926_

Round 1

Reviewer 1 Report

Comments and Suggestions for Authors

In this work, the authors Loi et al have presented a nice simulation on the 3D silicon sensor which is of great significance. This manuscript is well prepared, and I suggest the following publication as it is. Here are few suggestions:

1, The timing performance of 3D sensor showcases the optimized feature surpassing the counterparts of 1E, 2E. What are the difference of charge collection efficiencies between 3D and the others?

2, What will be influence (both positive and negative) of the angle-dependence charge collection on its following applications?

Author Response

Comment1: The timing performance of 3D sensor showcases the optimized feature surpassing the counterparts of 1E, 2E. What are the difference of charge collection efficiencies between 3D and the others?

Response1: If you consider only the active volume of all 3 geometries, charge collection efficiency is basically the same for all un-irradiated geometries due to their very short drift distance and, therefore, charge collection time, which is by far faster than the collection time and recombination and trapping time. On the other hand, the dead volume introduced by the trench sensor is larger than the columnar geometries, reducing overall detection efficiency.

Comment2: What will be influence (both positive and negative) of the angle-dependence charge collection on its following applications?
Response2: Tilting a 3D silicon sensor allows to increase detection efficiency because passing particles will not pass entirely through the electrodes that act like a dead volume which reduces the overall detection volume of the pixel. This allows to see most passing events. On the other hand, tilting reduces the released charge within the pixel as well as increases charge sharing among neighbouring pixel, redistributing the released charge among more pixel and, therefore, degrading the time resolution for each channel.

Reviewer 2 Report

Comments and Suggestions for Authors

The paper aims to optimize the timing performance of 3D silicon sensors with column electrodes with the express goal to reach a timing performance as close as possible to that of trenched electrode designs. Thus far, trenched designs have shown superior timing performance, but introduce concerns related to fabrication yield and pixel capacitance (impacting the noise). Both concerns could be mitigated by the more robust columnar electrode design. The paper presents a simulation study comparing different columnar designs to an established trench design, looking at weighting field, pixel capacitance, charge collection time, and charge distribution. A two-electrode (2E) design with 45 µm pitch is identified as the most promising variation to reach a timing performance close to that of 3D trenches.

The paper is well written, and the presented design study is comprehensive and clear. The presented results are of high interest to the community and potentially impactful as they establish a columnar 3D sensor design as a competitive alternative to the more challenging trenched design. As such, the paper presents an important step towards 4D tracking in the upcoming High Luminosity LHC era. I can, therefore, expressly recommend the paper for publication.

Below, I list minor points that could benefit the quality of the manuscript.

General:

- It would be interesting for readers if the authors could comment on the process parameters (in particular, doping concentrations) assumed for the simulated devices. Are the parameters the same as stated in reference [14] or have there been changes made? If there were changes, what was the reasoning behind? Even if no direct quotes can be made, it would be interesting to comment on changes with respect to previously tested and fabricated devices if there are any. In particular, it would be interesting to comment on whether the simulated 1E design is different to that of the devices tested in reference [8].

- In addition to the different geometry variations that are to be included in the upcoming fabrication at FBK, are there further process splits planned (e.g. doping concentrations)?

- While the 2E design shows approximately two times faster charge collection than the 1E design, the 1E design exhibits about two times lower capacitance. Could there be a comment made on the implications of this trade-off (e.g. implications for the readout electronics if the 2E design is chosen)?

Line-by-line comments:

- Some graphs would benefit from increased label and legend size: In particular, this pertains to Figure 3 (profile plots on the right), Figure 4 (legend, size, and placement), Figure 5 (all axis labels), Figure 8 and 9 (legend), Figure 10 (all labels).

- Figure 1: Adding a legend of the colour coding or labelling of the electrodes would improve readability.

- Line 79: “...might not be be maintained...” --> “...might not be maintained...”

- Line 118: “...compatible with existing and future readout chips...” For completeness, it would be interesting to mention and reference the concrete readout chips and technologies that are envisioned at this point in the text.

- Line 241: “dependent from” --> “dependent on”

- Line 301: “...positioned below of if.” --> “positioned below.”

- Figure 10 caption: “Please pay attention on...” --> “Please pay attention to..."

Author Response

Comment1:It would be interesting for readers if the authors could comment on the process parameters (in particular, doping concentrations) assumed for the simulated devices. Are the parameters the same as stated in reference [14] or have there been changes made? If there were changes, what was the reasoning behind? Even if no direct quotes can be made, it would be interesting to comment on changes with respect to previously tested and fabricated devices if there are any. In particular, it would be interesting to comment on whether the simulated 1E design is different to that of the devices tested in reference [8]. 

Response1:No, process parameters are the same as used in 14 regarding doping. The only difference is the way the DRIE is applied to achieve a trench electrode instead of a columnar electrode. About reference 8, the 1E device used in this study is based on a double face 3D technology, which digs the bias and readout electrodes from both sided of the wafer. Due to this approach, the wafer must be thicker to hold mechanical stability. The greater thickness is an advantage because the entire thickness of a double side 3D is all active volume but, at the same time, the capacitance increases due to the larger thickness. At the same time a double side 3D suffer more mask aligning issues due to the fact that front and back mask must be alighed separately, which in average causes mis-alignment of circa 3 to 5 micrometers.

Comment2: In addition to the different geometry variations that are to be included in the upcoming fabrication at FBK, are there further process splits planned (e.g. doping concentrations)?

Response2: At the moment I cannot answer on this question due to classified fabrication process strategy.

Comment3:  While the 2E design shows approximately two times faster charge collection than the 1E design, the 1E design exhibits about two times lower capacitance. Could there be a comment made on the implications of this trade-off (e.g. implications for the readout electronics if the 2E design is chosen)?

Response3:

It's true that the 1E presents a definitely lower capacitance compared to the
2E, which means a better signal to noise ration but the width of the charge collection time distribution dominates over the capacitance, influencing signal risetime distribution and, therefore intrinsic time resolution.

Comment3: Some graphs would benefit from increased label and legend size: In particular, this pertains to Figure 3 (profile plots on the right), Figure 4 (legend, size, and placement), Figure 5 (all axis labels), Figure 8 and 9 (legend), Figure 10 (all labels).
done

- Figure 1: Adding a legend of the colour coding or labelling of the electrodes would improve readability.
done
- Line 79: “...might not be be maintained...” --> “...might not be maintained...”
done
- Line 118: “...compatible with existing and future readout chips...” For completeness, it would be interesting to mention and reference the concrete readout chips and technologies that are envisioned at this point in the text.
the 55 micrometer pitch design is compatible with the entire medipix/timepix/velopix family. On the other hand the italian initiative IGNITE is developing a new, 45 um pitch ROC. A first comparison can be found on this reference, which I will add on the manuscript (https://www.sciencedirect.com/science/article/pii/S0168900224001645)
- Line 241: “dependent from” --> “dependent on”
done
- Line 301: “...positioned below of if.” --> “positioned below.”
done
- Figure 10 caption: “Please pay attention on...” --> “Please pay attention to..."
done

Reviewer 3 Report

Comments and Suggestions for Authors

The paper is interesting and well written, therefore it can be considered for publication after a few very minor revisions.

Please resize to enlarge as follows:

- 1D plots in fig.3, fig 10

- the legend in fig.4, fig 8 and fig 9

- 2D plots in fig. 5,  fig. 10.

line 58. Do you confirm that the timing resolution measurement was performed at 20ºC or it was at -20º, like in the other cases?

line 292-293. Please, can you comment the sentence "increase detection efficiency by avoiding complete release of charge within the electrodes"? It is not quite understandable for me.

Author Response

Comment 1: 
Please resize to enlarge as follows:
- 1D plots in fig.3, fig 10
- the legend in fig.4, fig 8 and fig 9
- 2D plots in fig. 5,  fig. 10.
done
line 58. Do you confirm that the timing resolution measurement was performed at 20ºC or it was at -20º, like in the other cases? Yes, it’s at -20 °. Compiling error, solved
line 292-293. Please, can you comment the sentence "increase detection efficiency by avoiding complete release of charge within the electrodes"? It is not quite understandable for me.
Basically, when the sensor is aligned with the direction of propagation of the particle, there might be the possibility that a particle passes completely through an electrode. The electrode, which is an empty volume/equipotential volume, is not sensible to charge release, reducing the active volume of the pixel. If you tilt the sensor, only a fraction of the path of the particle crosses through the electrode. The rest of the energy is released within the active volume, resulting therefore as a detectable event. 

OK, done.